



# Multi-dimensional, Multi-Constraint Seismic Inversion of Acoustic Impedance Using Fuzzy Clustering Concepts

Saber Jahanjooy[1], Hosein Hashemi[1], Majid Bagheri[1]

[1] Institute of Geophysics, University of Tehran, Tehran, Iran

*Correspondence to*: Hosein Hashemi (hashemy@ut.ac.ir)

**Abstract**

In the process of transforming seismic data into vital information about subsurface rock and fluid properties, seismic inversion is a crucial tool. This motivates researchers to develop several seismic inversion methods and software. Since the seismic data are band-limited, seismic inversion is ill-posed, and the results are not unique,

each method tries to use initial information and assumes expected conditions for the results. Satisfying a general low-frequency trend and having a smooth model or step-wise results are some of the assumptions that these methods add as constraints to the inversion process. Well-logs, geological studies, and models from other geophysical methods can add important insight into the seismic inversion results. We introduce an objective function that applies the clustering properties of the prior information as a constraint to the seismic inversion

process as well as other common constraints. An optimal solution to the objective function is explained. We applied the Gustafson-Kessel fuzzy C-means as one of the possible clustering methods for clustering term. Numerical synthetic and real data examples show the efficiency of the proposed method in the inversion of seismic data. In addition to the acoustic impedance model, the proposed seismic inversion method creates reliable deconvolved and denoised versions of the input seismic data. Additionally, the membership section output from

the inversion process shows high potential in the seismic interpretation. Further research on selecting an optimum fuzziness, updating wavelet, and the potential of the membership sections to track horizons, distinguish sequences and layers, identify possible contents of the layers, and other possible applications are recommended.



# 1 Introduction

Seismic sections are bandlimited data that manifest a general overview of the structural scheme of the subsurface
layers. However, post-processing steps and analysis reveal more comprehensive information on the geophysical
properties and contents of these layers. Seismic inversion creates models of subsurface properties such as acoustic
impedance from seismic traces. These models are the source of our information on petrophysical properties and
reservoir characterizations (Gogoi and Chatterjee, 2019; Jafri et al., 2017). The vital applications of these models
from seismic inversion in seismic interpretation and hydrocarbon exploration attract the attention of scientists to
develop effective and efficient seismic inversion methods.

Numerous seismic inversion methods have been introduced by researchers. These methods can be categorized
based on the processing step of the input seismic data (pre-stack and post-stack), the supporting data (Blind and
model-based), the mathematical calculations (deterministic and statistical, and the assumptions on the results. The
pre-stack seismic inversions use angle-stacked or offset-stacked seismic data and well-logs to create models of P-
impedance, S-Impedance, Vp/Vs, or variations of Lame's parameters (Mallick, 1995; Goodway et al., 1997).
These models could be used to interpret reservoirs, fluid contents, and rock properties (Vernik, 2016). However,
pre-stack inversion needs high-quality seismic data with a good signal-to-noise ratio and accurate AVO or AVA
analysis. It is also computationally expensive and time-consuming (Maurya and Singh, 2018). Post-stack seismic
inversions utilize the final stacked seismic sections to create a high-resolution acoustic impedance (AI) model of
the subsurface (Maurya et al., 2020). It estimates acoustic impedance which is related to lithology and porosity,
but not directly to fluid content. Post-stack inversion is simpler and faster than pre-stack inversion, and it can
produce higher-resolution images of the subsurface structures (Maurya and Singh, 2018).

Deterministic seismic inversions minimize the difference between seismic traces and synthetic traces from the
created model while satisfying predefined constraint(s) (Cooke and Cant, 2010). However, interpreting the results
of these methods could be difficult due to the different vertical scale than true geological models and well logs
and the limited frequency band of the result models (Francis, 2010). On the other hand, statistical inversions (Haas
and Dubrule, 1994; Bosch et al., 2010) use statistical properties of well-logs data and amplitudes of seismic data
to build a model on spatial grids. This procedure covers the limited frequency bandwidth of the seismic data by
using prior information. Another way to create broadband models is to use deterministic and statistical seismic
inversion simultaneously (Fu, 2004).

Single trace inversion creates simple interpretable results (Yilmaz, 2001). However, the non-realistic assumption
on density, band-limited, and low-resolution results make this method less applicable. Since seismic inversion is
an ill-posed problem (Tarantola, 2005) and has non-unique solutions, studies try to reduce the possible solutions
by adding constraints in the temporal-spatial domain or a transformation domain. For geophysical data and seismic
inversion, in which the interpreter expects some behaviors in the final model, different types of constraints are
commonly used. Tikhonov regularizations (Tikhonov et al., 1977) smooth the model (Li and Oldenburg, 1999),
while total variation (Rudin et al., 1992) decreases the effects of outliers and noises while detecting sharp boundary
changes. Blind seismic inversions add constraints to the inversion process and create models from seismic traces
(Jahanjooy et al., 2022). Model-based inversions start from a low-frequency model from well-logs and seismic
interpretations and create final models around the initial model (Cooke and Schneider, 1983)(She et al., 2019).
An inverse optimization problem by try and error of different objective functions is a straightforward method to
select an optimum objective function (Terekhov et al., 2010). Each of these constraints highlights one aspect of



the inversion model such as smoothness and ignores other aspects of the model such as step-wise transition of the layers.

Jointly inversion or parallel computing of other types of geophysical data is also used in creating geophysical models (Gyulai et al., 2013). This technique simultaneously calculates the geophysical models and overcomes the limitations of individual inversions by combining different types of data that are sensitive to different physical properties. As a result, this enhances the resolution and uniqueness of the inverted models and provides a more comprehensive interpretation (Zhdanov et al., 2012). Researchers have inverted seismic data with gravity,

magnetotelluric, and other types of geophysical data (Le et al., 2016; Bennington et al., 2015; Rapstine, 2015; Liao et al., 2022). However, joint inversion could be computationally expensive, and sensitive to noise and uncertainties in the prior information. Moreover, linking different geophysical data and models could be challenging.

Modern machine learning algorithms allow the incorporation of seismic data, well logs, and geological data to

create geophysical models (Barman and Sen, 2022; Liu, 2019). Machine learning seismic inversion aims to create models with higher resolution, faster convergence, better robustness, and less manual input (Chen and Saygin, 2021). However, challenges such as data quality, training data, and hyperparameter selection affect these methods (Meng et al., 2021).

In data science, the dimensionality and complexity of the data can be reduced by clustering. Similarly, geophysical

methods use clustering to treat the data (Hashemi et al., 2008) and extract useful features and patterns from the data (Moosavi et al., 2023). Moreover, it can impose structural or statistical constraints on the model, and enhance the resolution and robustness of the inversion results. In a seismic inversion, the most beneficial aspect of fuzzy clustering is that it can integrate prior non-seismic data into the inversion (Sun and Li, 2016). According to the Sun, (2016), nonlinearity, uncertainty, and heterogeneity of the subsurface structures and properties are handled

by fuzzy clustering terms in the inversion. Kieu and Kepic (2020) extend the model-based Occam's inversion (Cooke and Schneider, 1983) and add a fuzzy term to incorporate petrophysical information in the inversion (Kieu and Kepic, 2020). The extended method, which requires reliable borehole data and prior petrophysical constraints, may not be suitable for complex geological settings, where the seismic impedance is not well correlated with the petrophysical properties. The results are biased to the initial model and the choice of fuzzy clustering parameters.

This research proposes a new multi-term objective function for seismic inversion. The aim is to integrate the prior geophysical data from well logs, and seismic traces in the inversion process and deal with various constraints on the resulting model. The constraints include data fidelity, model perturbation, smoothness, sparsity, and clustering. Despite the complexity of the objective function, a simple solution by existing methods such as least squares or matching pursuit is suggested. Although the method is model-based, it has less dependence on the initial model

and wavelet. The proposed fuzzy seismic inversion creates a deconvolved reflectivity section, a denoised version of the seismic data, and membership sections, as well as the acoustic impedance model. The membership sections that show the degree of belonging of each data point to different rock types or facies are new concepts that could be used in seismic interpretation.





## 2 Theory

### 2.1 Data misfit

The physical properties of the subsurface are unknown, and the goal is to estimate these properties, from limited geophysical data through ill-posed optimizations. For the reflection seismic traces, $d$ is the convolution result of seismic wavelets, $w$ and reflectivity series, $r$ ($d = w * r$). If $\mathbf{W}$ is the convolution matrix of the wavelet then the seismic trace(s) are a multiplication of $\mathbf{W}$ and $r$ ($d = \mathbf{W}r$). In other words, the data misfit is a linear relation of the trace(s) and reflectivity:

$$\boldsymbol{\Phi}_d = \|d - \mathbf{W}r\|_2^2.$$

(1)

where $\|.\|_2^2$ is the Frobenius (Euclidean) norm. Berteussen and Ursin (1983) have formulated the reflectivity for a continuous acoustic impedance media ($\mathbf{z}$) (Berteussen and Ursin, 1983):

$$r_j = \frac{1}{2}\ln\left(\frac{z_{j+1}}{z_j}\right) = \frac{1}{2}\log(z_{j+1}) - \frac{1}{2}\log(z_j) = x_{j+1} - x_j. \quad (a)$$

$$r = D_v x. \quad (b)$$

(2)

where $Z_j$ is the acoustic impedance of the $j^{th}$ sample point, and $D_v$ is the first-order differential matrix along the time dimension. Conversely, having $r$, the value of x can be obtained in a recursive calculation using Eq. (3):

$$x_{j+1} - x_j = 2\sum_{i=1}^{j} r_i. \quad (a)$$

$$x = Hr. \quad (b)$$

$$z_j = z_1 e^{2\sum_{i=1}^{J} r_j} = z_1 e^{Hr}. \quad (c)$$

$$r = [r_1. r_2. \dots. r_N]^T. \quad H = \begin{bmatrix} 0 & 0 & \cdots & 0 \\ 2 & 0 & \cdots & 0 \\ 2 & 2 & \cdots & 0 \\ \vdots & \vdots & \ddots & \vdots \\ 2 & 2 & \cdots & 2 \end{bmatrix}.$$

(3)

### 2.2 Model perturbation

An initial guess could improve the uniqueness of the inversion model (Mallick, 1995). This guess usually is based on the well-logs and contains the general low-frequency trend of the subsurface model ($x^{(0)}$). The model-based inversion methods, the final model is a small perturbation around the initial modal:

$$\boldsymbol{\Phi}_x = \left\|x^{(0)} - Hr\right\|_2^2.$$

(4)




### 2.3 Sparsity

Another constraint to reduce the ill-posedness of the seismic inversion problem is the sparsity of the reflectivity.
Adding a sparsity constraint, $\boldsymbol{\Phi}_r$ on the reflectivity facilitates the detection of the high-frequency variations and

thin-beds:

$$\boldsymbol{\Phi}_r = |\boldsymbol{r}|$$

(5)

### 2.4 Smoothness

In cases of noisy data and when the model is expected to be consistent, a smoothness term, $\boldsymbol{\Phi}_s$ is a logical penalty

for optimizing the result. Considering the dimensions of the data and the model properties, the penalty term is
defined as either the first or second derivative of the model along each dimension or a combination of these
derivatives (Constable et al., 1987). Equation ( 6 ) lists a collection of smoothness objective functions that do not
conflict with the sparsity of the reflectivity. $\boldsymbol{D}_h$ is the first-order differential matrix along the spatial dimension.

Where it is expected that the model has less variation in the horizontal direction, $\boldsymbol{\alpha}$ is a small factor that controls

the smoothness variation.

$$
\begin{aligned}
&\boldsymbol{\Phi}_s = \|\boldsymbol{D}_h \boldsymbol{x}\|_2^2 = \|\boldsymbol{D}_h \boldsymbol{H} \boldsymbol{r}\|_2^2. \quad (a)\\
&\boldsymbol{\Phi}_s = \|\boldsymbol{D}_h^2 \boldsymbol{x}\|_2^2 = \|\boldsymbol{D}_h^2 \boldsymbol{H} \boldsymbol{r}\|_2^2. \quad (b)\\
&\boldsymbol{\Phi}_s = \|\boldsymbol{D}_v \boldsymbol{D}_h \boldsymbol{x}\|_2^2 = \|\boldsymbol{D}_v \boldsymbol{D}_h \boldsymbol{H} \boldsymbol{r}\|_2^2 = \|\boldsymbol{D}_h \boldsymbol{r}\|_2^2. \quad (c)\\
&\boldsymbol{\Phi}_s = \|\alpha \boldsymbol{D}_v \boldsymbol{x}\|_2^2 + \|\boldsymbol{D}_h \boldsymbol{x}\|_2^2 = \|\alpha \boldsymbol{D}_v \boldsymbol{H} \boldsymbol{r}\|_2^2 + \|\boldsymbol{D}_h \boldsymbol{H} \boldsymbol{r}\|_2^2 = \left\| \begin{pmatrix} \alpha \boldsymbol{D}_v \\ \boldsymbol{D}_h \end{pmatrix} \boldsymbol{H} \boldsymbol{r} \right\|_2^2. \quad (d)\\
&\boldsymbol{\Phi}_s = \|\alpha \boldsymbol{D}_v^2 \boldsymbol{x}\|_2^2 + \|\boldsymbol{D}_h^2 \boldsymbol{x}\|_2^2 = \|\alpha \boldsymbol{D}_v^2 \boldsymbol{H} \boldsymbol{r}\|_2^2 + \|\boldsymbol{D}_h^2 \boldsymbol{H} \boldsymbol{r}\|_2^2 = \left\| \begin{pmatrix} \alpha \boldsymbol{D}_v^2 \\ \boldsymbol{D}_h^2 \end{pmatrix} \boldsymbol{H} \boldsymbol{r} \right\|_2^2. \quad (e)\\
&\boldsymbol{\Phi}_s = \|\alpha \boldsymbol{D}_v \boldsymbol{x} + \boldsymbol{D}_h \boldsymbol{x}\|_2^2 = \|(\alpha \boldsymbol{D}_v + \boldsymbol{D}_h) \boldsymbol{H} \boldsymbol{r}\|_2^2. \quad (f)\\
&\boldsymbol{\Phi}_s = \|\alpha \boldsymbol{D}_v^2 \boldsymbol{x} + \boldsymbol{D}_h^2 \boldsymbol{x}\|_2^2 = \|(\alpha \boldsymbol{D}_v^2 + \boldsymbol{D}_h^2) \boldsymbol{H} \boldsymbol{r}\|_2^2. \quad (g)
\end{aligned}
$$

(6)

To extend the algorithm to three dimensions, instead of $\boldsymbol{D}_h$ in the smoothness term, one can simply use $\boldsymbol{D}_x$ and

$\boldsymbol{D}_y$, which are the first difference matrices along the inline and crossline directions.

### 2.5 Clustering

Generally, a subsurface geological medium consists of different segments, sequences, or layers. Clustering assigns
similar points of the subsurface model to distinct groups. Hard clustering methods such as K-means assume that
each data point belongs to an exclusive group that has a calculated cluster center or centroid (Han and Kamber,

2006).  For a set of data points, $x_j$ and centroids, $o_k$, K-means minimizes the objective function in Eq. ( 7 ):

$$\Phi_{c\,(KM)} = \sum_{j=1}^{M} \sum_{k=1}^{C} v_{jk} d_{jk}^2$$

(7)

where $\boldsymbol{d}_{jk}$ is the distance between a data point and its respective centroid. $\boldsymbol{v}_{jk}$ is 1 if $\boldsymbol{x}_j$ belongs to cluster k, and 0
otherwise.

Unlike hard clustering, which allows data points to belong to only one group, soft or fuzzy clustering allows data
points to belong to multiple groups with relative memberships. A fuzzy clustering method such as Fuzzy C-Means,



calculates the centroids $\boldsymbol{o}_k$ of the clusters and the normalized memberships $\boldsymbol{u}_{jk}$ of each data point to these centroids in an iterative process. As in the K-means, FCM aims to minimize the objective function in Eq. ( 8 ) (Bezdek, 2013):

$$\boldsymbol{\Phi}_{c\,(FCM)} = \sum_{j=1}^{M}\sum_{k=1}^{C} \boldsymbol{u}_{jk}^{q}\boldsymbol{d}_{jk}^{2}.$$

( 8 )

where q is a parameter that indicates the fuzziness of the data. Depending on the primary assumptions about the shapes of the clusters, distances can be in various forms such as Euclidian ( Eq. ( 9 )) as in FCM or Mahalanobis (Mahalanobis, 1936) as in Gustafson Kessel (GK) fuzzy clustering (Krishnapuram and Kim, 1999).

$$\boldsymbol{d}_{jk}^{2} = \left\| \boldsymbol{x}_j - \boldsymbol{o}_k \right\|_{2}^{2}.$$

( 9 )

The clustering objective function in the Eq. ( 8 ) has two variables, $\boldsymbol{u}$, and $\boldsymbol{O}$. To solve for each variable, one needs to differentiate the objective function for one of these variables while the other variable is treated as known and set the resulting expressions to zero (Zadeh, 1965). Starting with an initial guess for centroids, the centroids and 175 memberships are updated in an iterative process:

$$\boldsymbol{u}_{jk} = \frac{1}{\sum_{i=1}^{C}\left(\dfrac{\boldsymbol{d}_{jk}}{\boldsymbol{d}_{ji}}\right)^{\frac{2}{q-1}}}. \quad (a)$$

$$\boldsymbol{O}_k = \frac{\sum_{j=1}^{M}\boldsymbol{u}_{jk}^{q}\,xj}{\sum_{j=1}^{M}\boldsymbol{u}_{jk}^{q}}. \quad (b)$$

( 10 )

Using proper distance calculation modifies the clustering results to the behavior of the input data (Krishnapuram 180 and Kim, 1999). On the other hand, spatial properties such as stepwise or smooth transition within each cluster are addressed in several fuzzy clustering research. In the revised fuzzy clustering methods, spatial information is incorporated into the clustering process (Pham, 2001; Li et al., 2011; Chuang et al., 2006). To account for the probable correlation of the data points to their neighboring, Chuang et al (2006) add the local behavior of the clusters ($h$) in windowed areas around the calculated data point (Chuang et al., 2006):

$$h_{jk} = \sum_{l \in L} \boldsymbol{u}_{lk}$$

( 11 )

Where L is the indices collection for the neighboring window $\boldsymbol{x}_j$. In this case, the membership will be updated by using Eq. ( 12 ) where the $\beta$ and $\gamma$ parameters control the contribution of $\boldsymbol{u}$ and $\boldsymbol{h}$. Knowing that the $u_j$ is normalized, when $\beta = 1$ and $\gamma = 0$, $u'_{jk}$ is equal to $\boldsymbol{u}_{jk}$.




$$u'_{jk} = \frac{\boldsymbol{u}_{jk}^{\beta} \boldsymbol{h}_{jk}^{\gamma}}{\sum_{i=1}^{C} \boldsymbol{u}_{ji}^{\beta} \boldsymbol{h}_{ji}^{\gamma}}$$

( 12 )

A main concern in the clustering process is the number of clusters. Elbow methods are widely used for determining the optimum number of clusters (Bezdek and Pal, 1998). Here, well-data are used to estimate the cluster number. Although this information gives valuable insight into the overall model, the problem arises when

the clusters of the well-site do not exist in the other location or conversely, there are additional clusters in the subsurface. The later problem could also arise when windowing the seismic data to smaller sub-sections. In this particular scenario, there may be a smaller number of clusters in each window. The large numbers of pre-defined clusters overlap, and/or there are clusters in which all of their memberships are relatively small. To address this challenge and avoid the use of the elbow method for each windowed data, a Gram matrix is created (Horn and

Johnson, 2012):

$$G = \boldsymbol{u}^T \boldsymbol{u}$$

( 13 )

Data points have small memberships to the centroids which are far from the window data. Therefore, the diagonal elements related to these clusters in the Gram matrix are small. On the other hand, the centroids which have high

overlap, create large non-diagonal elements. Omitting the highly overlapped clusters and the clusters which have negligible memberships, optimizes the number of clusters in the windowed data.

**2.6 Seismic inversion**

Equation ( 1 ) has non-unique answers. Adding a model perturbation term ($\boldsymbol{\Phi}_x$) keeps the resulting model close to the initial low-frequency guess, while the sparsity term ($\boldsymbol{\Phi}_r$) tries to enhance the resolution of the data. The

smoothness term $\boldsymbol{\Phi}_s$ considers the spatial-temporal properties of the resulting model. A clustering term $\boldsymbol{\Phi}_c$ forces the model to satisfy initial information on the model. This information could come from well-logs, seismic interpretation, and/or geological settings. To decrease ambiguities of the result, all of the mentioned terms are considered and a multi-term objective function is created:

$$r = argmin_r\{\boldsymbol{\Phi}\} = argmin_r \left\{\frac{w_d \boldsymbol{\Phi}_d}{\theta(d)} + \frac{w_x \boldsymbol{\Phi}_x}{\theta(x)} + \frac{w_s \boldsymbol{\Phi}_s}{\theta(c)} + \frac{w_c \boldsymbol{\Phi}_c}{\theta(c)} + \boldsymbol{\Phi}_r\right\}.$$

$$\theta(d) = \frac{norm(\boldsymbol{d}.2)}{length(\boldsymbol{d})}. \quad \theta(x) = \frac{norm(\boldsymbol{x}.2)}{length(\boldsymbol{x})}. \quad \theta(c) = \frac{norm(\boldsymbol{u}^q \hat{O}.2)}{length(\boldsymbol{x}) \times C}$$

( 14 )

Each objective term is weighted using a denominator, $\theta(\cdot)$ to normalize contribution of the terms. The $norm(\boldsymbol{d}.2)$ is the Euclidian norm. $\hat{O}$ is a stack of identical copies of O with its size equal to the memberships, $\boldsymbol{u}$. Note that the $\theta$ is the same for the smoothing and clustering terms. Weighting factors balance the misfit or data

fidelity term and the other terms (such that $w_d + w_x + w_s + w_c = 1$). This eliminates the need for high dimensional L-curve (Brooks et al., 1999), L-surface (Brooks et al., 1999), or Generalized cross-validation (Golub et al., 1979) methods to find optimum regularization parameters.

Various forms of hard or soft clustering may be employed as the clustering term. To solve Eq. ( 14 ), $\boldsymbol{\Phi}$ is rearranged as a conventional L2-L1 problem in the Eq. ( 15 ). Common methods such as the iterative reweighted



least square (IRLS) or the orthogonal matching pursuit (OMP) may be employed to solve this new objective
function.

$$\boldsymbol{\Phi} = \|\mathbf{Ar} - \boldsymbol{b}\|_2^2 + |\boldsymbol{r}|.$$

$$\widehat{\boldsymbol{d}} = \begin{bmatrix} d_1 \\ d_2 \\ \vdots \\ d_N \end{bmatrix}, \quad \widehat{\boldsymbol{x}}_0 = \begin{bmatrix} x_1 \\ x_2 \\ \vdots \\ x_M \end{bmatrix}, \quad \hat{I} = \begin{bmatrix} 0 \\ 0 \\ \vdots \\ 0 \end{bmatrix}, \quad \widehat{U} = \begin{bmatrix} diag\{\sqrt{\boldsymbol{u}_1}\} \\ diag\{\sqrt{\boldsymbol{u}_2}\} \\ \vdots \\ diag\{\sqrt{\boldsymbol{u}_C}\} \end{bmatrix}, \quad \widehat{UO} = \begin{bmatrix} diag\{\sqrt{\boldsymbol{u}_1}o_1\} \\ diag\{\sqrt{\boldsymbol{u}_2}o_2\} \\ \vdots \\ diag\{\sqrt{\boldsymbol{u}_C}o_C\} \end{bmatrix}.$$

$$\widehat{\boldsymbol{W}} = \begin{bmatrix} W & & & \\ & W & & \\ & & \ddots & \\ & & & W \end{bmatrix}, \quad \widehat{\boldsymbol{H}} = \begin{bmatrix} H & & & \\ & H & & \\ & & \ddots & \\ & & & H \end{bmatrix}, \quad \boldsymbol{A} = \begin{bmatrix} w_1\widehat{W} \\ w_2\widehat{H} \\ w_4\widehat{D} \\ \sqrt{w_5}\widehat{U}\widehat{H} \end{bmatrix}, \quad \boldsymbol{b} = \begin{bmatrix} w_1\widehat{d} \\ w_2\widehat{x}_0 \\ \hat{I} \\ \sqrt{w_5}\widehat{U}O \end{bmatrix},$$

230 (15)

Having seismic data, wavelet, initial model, and prior clustering information, Eq. ( 15 ) solves for reflectivity.
Equation ( 3 ) uses this reflectivity and creates an impedance model.

Solving Eq. ( 10 ) for memberships and centroids and repeating the inversion converges the model, the centroids,
and the memberships. Note that in the existing methods of clustering, centroids are updated iteratively until they

converge. However, in each iteration of the seismic inversion, they update only ones. The interpreter can include
the time/depth of the cluster in the clustering step ($[\boldsymbol{t}.\boldsymbol{x}]$) and divide the model into a pre-defined number of
overlapped regions, or only applies cluster magnitude of the model ($[\boldsymbol{x}]$).





## 3 Application Examples


The proposed method is applied to a synthetic seismic section and an inline section of the F3 block data set. The basic structure of the synthetic model allows us to observe the results in common 1D and 2D cases. The complex structures in the F3 data and the presence of hydrocarbons in the North Sea make the inversion process more challenging.

### 3.1 Synthetic example


Figure 1a shows a synthetic geological model that consists of six major geological segments. A seismic section is created by convolving a Ricker wavelet and the reflectivity of the synthetic model. The signal-to-noise ratio of the created data with a 30 Hz dominant frequency is 1dB (Figure 1b). The true model is smoothed along vertical and lateral directions (Figure 1c) and the result is used as an initial model.


Prior information such as the number and quantities of the main model clusters are usually attainable using clustering of the well-data and well-logs. However, for the synthetic model, the segment's AI is available and the cluster number, C is equal to 4. The proposed objective function could employ different clustering approaches ($\Phi_c$) as well as various smoothness terms ($\Phi_s$). **Figure 2** shows normalized mean square errors (NMSE) for the results of the inversion, while K-means and FCM are used as the clustering term. 1D stands for when each trace


is inverted separately. The second case is when the smoothness term in two-dimensional data is not used. For the 2D cases, in which neighbouring traces are considered, the calculation bins are windows of three traces. Although the input seismic data has high noise level, both clustering approaches have relatively small errors. However, the NMSE of fuzzy-based inversions are smaller. As an example, Figure 3 shows the resulting model when the clustering term is fuzzy and the smoothness is as Eq. (7b). Despite the high noise level, the main cluster regions


are separated.

Model-based inversions converge around the low-frequency initial models. Errors in the initial model remain in the final result. However, forcing the result to obey the clustering term moves the model toward its true value. Figure 3c displays an inversion result which, compared to Figure 3b, is started from a smoother and smaller initial model. However, the inversion result converged toward the true model.


Data fidelity term ($\Phi_d$) uses a known wavelet to estimate the seismic response of the reflectivity series. Incorrect wavelets lead to miscalculations and incorrect models. To check the dependence of the inversion results on the wavelet, the used Ricker zero phase wavelet is rotated. One trace of the synthetic section is inverted using different rotated Ricker wavelets. NMSE of the results are presented in Figure 4. Compared to the true zero-phase wavelet, a wide range of incorrect rotated wavelets create models which have NMSE comparable to the true wavelet.


Due to the low sensitivity of the proposed method to the noise level and the wavelet, the deconvolved traces are reliable. This could be seen in Figure 5 where the resulting reflectivity of trace 100 is compared to the true reflectivity. The reflectivity of the main boundaries is recovered. Although some small random artificial spikes are created, this does not affect the lateral continuity of the layers (Figure 3a). Consequently, the synthetic trace created by using the resulting reflectivity is significantly denoised (Figure 5).



### 3.2 Real case example

A Seismic inline section of the Netherlands OpendTect F3 dataset which intersects F034 well is presented in Figure 6a. Five of the main seismic horizons are interpreted. The elbow method for Gustafson-Kessel clustering of impedance well log suggests 11 clusters in the sampled depth. However, the seismic horizons show six general facies segments (Chevitarese et al., 2018) and some sublayers. A seismic wavelet is created by statistical wavelet estimation. The initial low-frequency model is created using Hampson-Russel software and the inversion bins are windows of three traces.

Four different scenarios are tested. In the first scenario, the initial model and centroids are available. Usually, well-logs provide some general properties of the model. Otherwise, the initial model can be created by using interpreted horizons, migration velocity, or other geophysical data. In the second scenario, assuming no clustering information, the initial model is clustered to create initial centroids for the inversion process. In a less probable case, for the third scenario, the initial model is not available or it is not created accurately but there is trusted initial clustering information. In this case, the inversion starts without the model perturbation term. The resulting model is used as the initial model for the next iterations and it updates at each iteration. The fourth scenario assumes no initial model and clustering information. In this case, the inversion starts without the model perturbation and clustering terms. The resulting model is the starting initial model and clustering source for the next iteration, which uses all of the objective terms. In reality, the initial model and clustering of an inversion bin can be used as initial information for the neighbouring bins.

On the other hand, in the inversion iterations, four different approaches can be used to update the centroids. In the first approach, the centroids only have one dimension, AI. While the magnitudes of the centroids are fixed, the memberships update at each iteration. The second approach also considers the time of the fixed centroids. In the third approach, the one-dimensional AI of the centroids and their memberships update at each iteration. The fourth approach updates both the magnitude and time of the centroids.

The proposed method, when it uses Ed. (7d) as smoothness, and C is equal to 6, is applied to the seismic section. The centroids are calculated using sonic and density logs at the location of the crossline 1008. As in approach two, only the memberships are updated. Although it is not a vital requirement, to reduce the number of iterations, it is recommended to start the inversion from the well-site location and use the memberships of the previous bin as a trusted initial membership for the neighbouring bins. While the fuzziness, q is equal to 2, the objective function converges after three iterations, ,Figure 6b and Figure 6c display the inversion results of the proposed method and model-based inversion of Hampson-Russel (HR) software, respectively. Although the models are analogous, the MFSI model has higher vertical and horizontal resolution and creates higher lateral continuity of the events.

The AI log derived from the density and sonic logs of the F034 well, F3 Block is displayed in Figure 7. Due to an error in the time-depth correction of the well-logs, there is an inconsistency from 1.5s to the end. A neighbouring trace to this well, which is at the intersection of inline 441 and crossline 1008, is compared to the synthetic seismogram that is obtained by convolving the wavelet and reflectivity of the resulting inversion model. The synthetic seismogram matches well with the original seismic trace and the error is relatively small.

In addition to the AI model, the inversion creates membership of each model point to each cluster center. Membership sections are sections with the dimensionality of the model in which each data point is the membership to one of the centroids. If there are C clusters, C separate membership sections will be created in which each





section is a display of belonging of the model points to one of the cluster centers. These membership sections help to track and interpret the data. The membership sections created in the inversion process of Figure 6a, are displayed in the top row of Figure 8. Note that for all the images the colour range is 0 to 1. For simplicity of the figure plots, the axis of these sections is not shown here. As they match the interpreted horizons in Figure 6a, these membership sections can be used to interpret and track horizons. The main sequences are distinguished.

Furthermore, the within-boundary variations of the model can be perceived and comprehended. The fuzziness, q in the clustering term is set to 1.1 and the inversion is repeated. The result membership sections are displayed in the lower row of Figure 8. As is expected from the fuzzy clustering concept, the lower fuzziness tends the fuzzy clustering toward hard clustering and decreases the resolution of the membership sections.

        As examples of all the possible combinations of different scenarios and different approaches, all the approaches
in the first scenario and the fourth approach in the other scenarios are applied in the different inversion process. The result membership sections comparable to the third membership in Figure 8 are displayed in Figure 9. The first approach (Figure 9a) unveils a detailed insight into the distribution of model magnitude in the subsurface. The second approach (Figure 9b) creates smoother cluster centers that track the fine impedance zones. However, the third approach creates more continuous and sharp results (Figure 9c). As in approach two, the fourth approach
(Figure 9d) distinguishes sequences or layers of the subsurface. Nevertheless, it enhances the imprecise initial cluster data and has more lateral continuity. Figure 9e, f, and g display the membership section resulting from approaches two to four respectively while updating AI and times of the centroids. Although these approaches do not have all or part of the initial information, they create results comparable to the first approaches with less resolution.

In a different calculation, the elbow method on the well-log estimates 11 clusters as the optimum cluster number and creates centroids. The result membership sections of the inversion process are displayed in Figure 10. Although finer variations of the memberships are created, in some regions the membership sections have a high correlation with other membership sections.





## 3 Discussion

The results of ill-posed problems such as seismic inversion greatly benefit from initial knowledge of the result. Adding additional constraints on the result decreases the ambiguity of the results and increases their uniqueness. The proposed MFSI algorithm aims to add multiple constraints to the inversion process. As a vital term of the

MFSI, fuzzy clustering of known data influences the inversion result. The objective function of the MFSI has five terms and solving it using traditional ways can be challenging. Empirically, weighting parameters have resolved the need to calculate multiple regularization parameters (Eq. ( 14 )). Equation ( 15 ) simplifies the multi-term objective function to a common norm one problem.

The examples (Figure 3) show that the data fidelity term of the proposed fuzzy seismic inversion ensures that the

synthetic seismic data from the resulting model is close to the input seismic data. The model perturbation term is a guarantee that the resulting model obeys a provided low-frequency model of the subsurface. This could convey the possible errors in the initial model to the resulting model. Depending on the expected behaviour of the subsurface and to account for lateral or vertical variations in the subsurface, the different provided smoothness terms could be examined. The sparsity constraint on the seismic reflectivity is an effort to achieve high vertical

resolution. Usually, well-log data, geological sections of the subsurface, other geophysical methods, or seismic interpretation create a general view of the subsurface sequences or layers which could be used as prior information through the use of the clustering term.

The multi-objectiveness of the MFSI reduces the dependency of the results on the errors in the initial data (Figure 3 and Figure 4). MFSI uses a fixed wavelet in all of its iterations. Although the algorithm allows small errors in

the wavelet, further research to update the wavelet at each iteration can improve the result. For situations such as non-stationary wavelets along time direction, and variation of the wavelet along spatial direction, the mentioned low-affection of the result by the wavelet is important. Some researchers propose to use the time-varying wavelet in the seismic inversion and improve the result  (Van Der Baan, 2008; Zhang and Fomel, 2017).

The MFSI creates membership sections, as well as acoustic impedance sections (Figure 8). Based on the available

data and assumptions on the cluster centers, one can compute the centroids and membership sections in four different scenarios, each using one of four possible approaches. The flexibility of the memberships sections (Figure 9 and Figure 10) makes them a valuable source for the interpreters and researchers to track horizons, distinguish sequences and layers, and identify possible contents of the layers. Here, the fuzziness is selected as 2. Decreasing the fuzziness shifts the fuzzy clustering toward hard clustering (Figure 8). However, the MFSI would

benefit from further research on selecting the optimal fuzziness in its clustering term.


## 4 Conclusion

This research introduces a multi-objective fuzzy seismic inversion method based on different constraints to create
an impedance model from post-stack seismic data. The proposed objective function is a combination of five well-known objective functions, each of which ensures a desired property in the resulting model. A simple solution to the new objective function is presented. Numerical synthetic and real data examples demonstrate the effectiveness of the proposed method in inverting seismic data. Although the method is model-based, the numerous constraints on the result make the proposed method less dependent on the initial data. Therefore, the reflectivity section
generated by the algorithm is a reliable deconvolved result and it can create a denoised version of the input seismic data. The clustering term is a vital part of the inversion process. Examples showed that in this form of the proposed method, fuzzy clustering creates more reliable results than hard clustering. The resulting membership sections of the inversion help the interpreters track horizons, distinguish sequences and layers, and identify possible contents of the layers. Further research on the initial wavelet, the optimal fuzziness, and the applications of membership
sections is recommended. In summary, this research proposes a novel and robust seismic inversion method that can produce high-quality impedance models with various desirable features and facilitate the interpretation of seismic data.





### Data Availability

The real seismic data used in this study is from the F3-Demo-2020 dataset, which is publicly available at TerraNubis (https://terranubis.com/datainfo/F3-Demo-2020).

### Competing Interests

The contact author has declared that none of the authors has any competing interests.




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


**Figures**

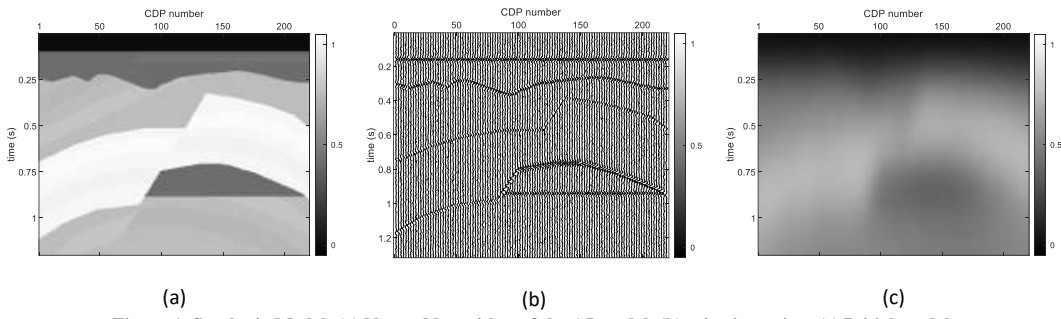

**Figure 1. Synthetic Model. (a) Natural logarithm of the AI model. (b) seismic section. (c) Initial model.**


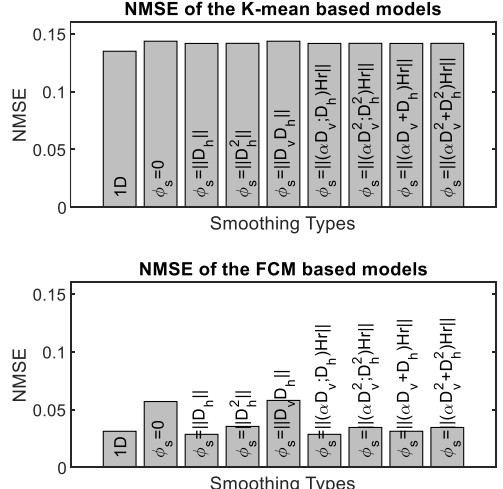

**Figure 2. Normalized mean square error of the result models using different smoothness terms. K-means is used as the fuzzy term (upper plot). FCM is used as the fuzzy term (lower plot).**

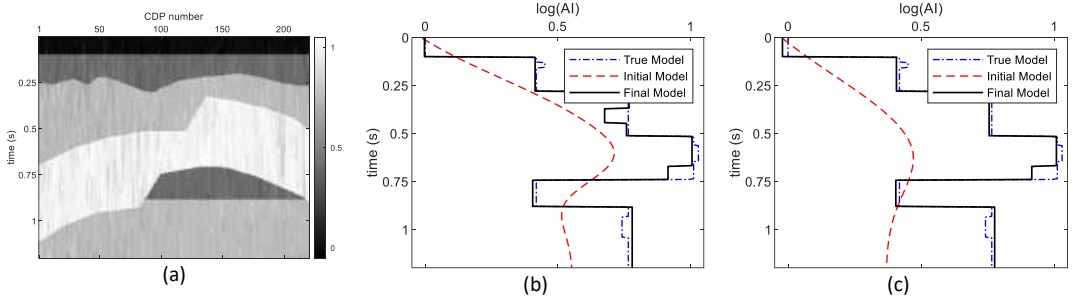

**Figure 3. Calculated AI models using the first difference matrix and prior information. (a) (b). (c) a log at the location of trace 100.**

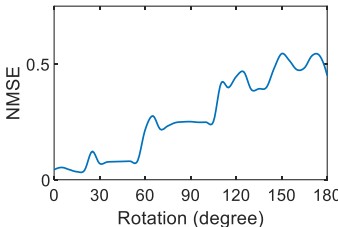

**Figure 4. NMSE of the resulting model for trace 100 using incorrect phase rotation of the Ricker wavelet.**


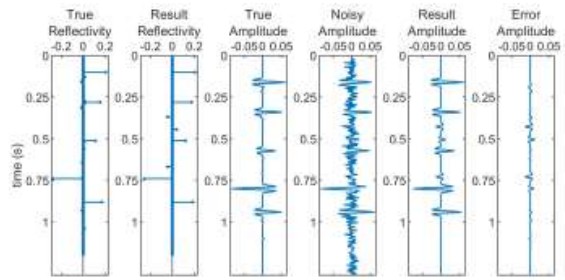

**Figure 5. True and calculated data at the location of CMP 100 of the synthetic data.**

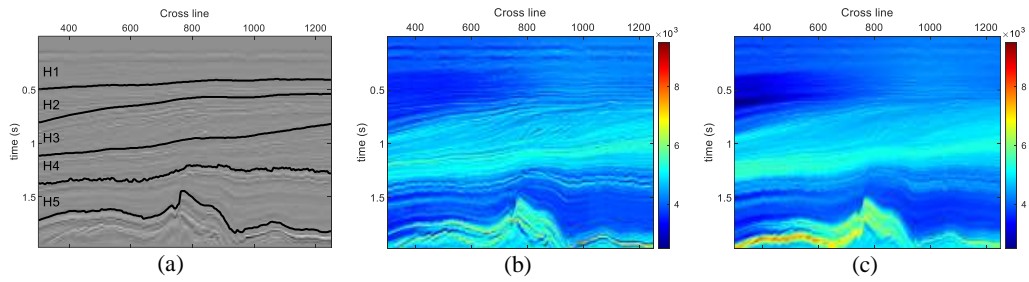

**Figure 6. Seismic section of Inline 441, F3 Block and five interpreted horizons (b) Inversion result of the third approach of the proposed method. (c) Model-based inversion.**

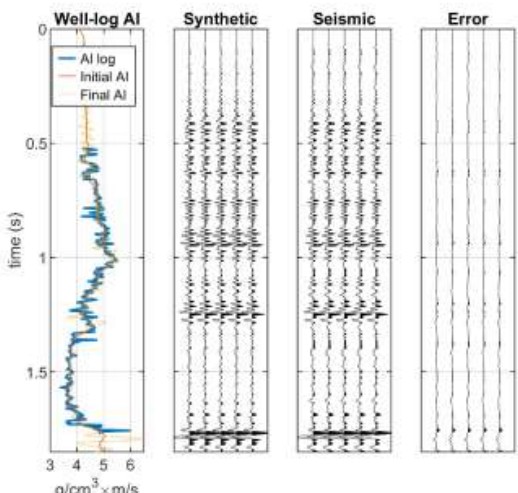

**Figure 7. Acoustic Impedance at the intersection of inline 441 and crossline 1008 in the location of the well F034, F3 Block data. The synthetic seismogram from the inversion result is compatible with the original seismic trace and the error is relatively small.**


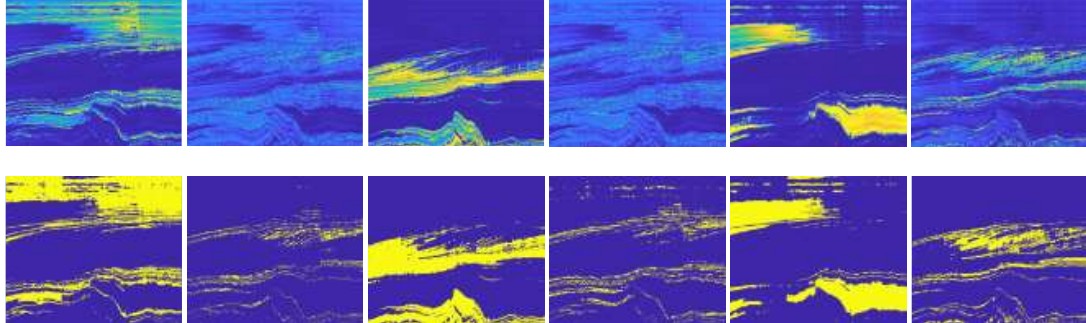

**Figure 8. Membership sections of Figure 6. b in which q=2 (upper row). Membership sections of the same inversion while q=1.1(lower row).**

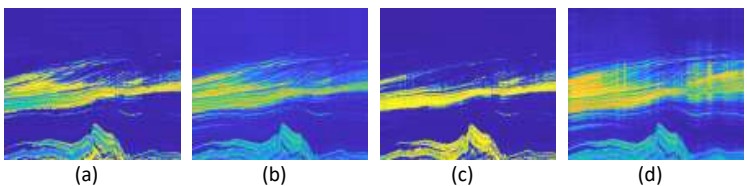

(a)    (b)    (c)    (d)

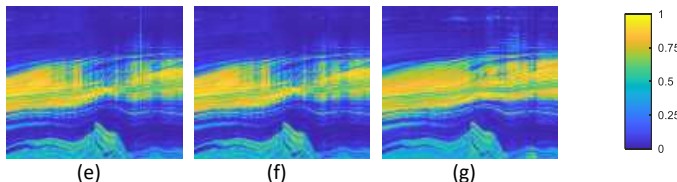

(e)  (f)  (g)

Figure 9. Membership sections comparable to Figure 7 in different situations of initial data and updating clusters. (a)
Initial model and clusters, fixed AI of the centroids. (b) Initial model and clusters, fixed AI, and times of the
centroids. (c) Initial model and clusters, updating 1D AI of the centroids. (d) Initial model and clusters, updating AI
and times of the centroids. (e) Initial model, no initial clusters, updating AI and times of the centroids. (f) No initial
model, initial clusters, updating AI, and times of the centroids. (g) No initial model, no initial clusters, updating AI
and times of the centroids.

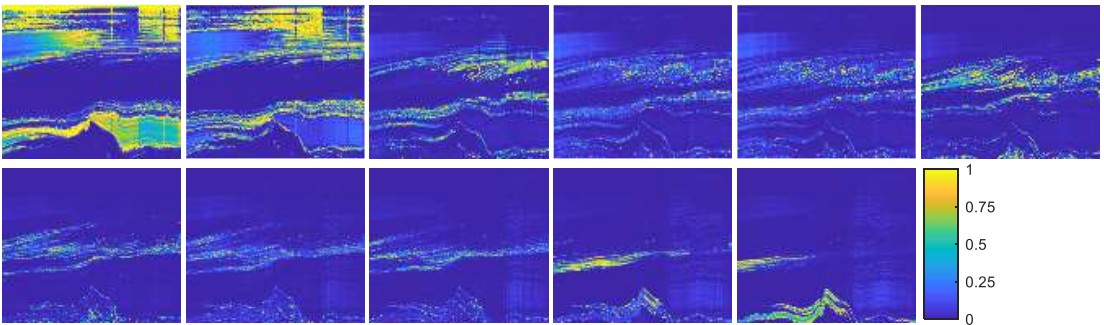

Figure 10. All membership sections of the inversion result when the cluster numbers are 11.