# Peer review of "Multi-dimensional, Multi-Constraint Seismic Inversion of Acoustic Impedance Using Fuzzy Clustering Concepts"

_Nonlinear Processes in Geophysics, 2024_

## Referee Comment (RC1)

[referee-annotated manuscript omitted]

---

## Referee Comment (RC2)

Dear Authors,

Thank you for your responses and explanations. While I agree with some of your points, overall, I am not satisfied with the responses. I would like to address three key aspects: the format of the responses, the answers to my main questions, and the comments in the text.

1. The form of the answers.
   I could not locate an updated version of your manuscript; I only see the old file with your responses. It is difficult to follow your answers without the use of track changes. Please ensure that the updated manuscript with track changes is provided for easier review.
2. The answers of my "main concerns"
- Objective Function Parameters (Equation 14):

You stated that the weighting factors ($w_d$, $w_x$, $w_s$, $w_C$) either have equal contributions or are adjusted based on initial model information.

My concern is that if these parameters have no effect when equal ("equal contributions"), how does the operator determine their weighting values?

Could you please provide a clearer explanation of how prior information (e.g., geology, boreholes) is incorporated into the inversion process?

I suggest adding more detailed statements, discussion, and supporting evidence to the text.

- Clustering Parameters:

The parameters related to clustering include the "Gustafson-Kessel fuzzy C-means" method, fuzziness, and the number of clusters.

- Your theory section discusses conventional Fuzzy C-means (FCM), but there is no clear distinction between FCM and the Gustafson-Kessel method in relation to your inversions.
- The theory section lacks a discussion of fuzziness and the number of clusters.
- Although spatial properties are mentioned, I do not see them addressed in the inversion process.

Please clarify these points and provide relevant discussions and evidence in the manuscript.

- Inversion Results with Different Noise Levels and Starting Models:

I believe adding results for different noise levels and initial models would not require much extra space if you manage the presentation more concisely and reduce the number of cases.

Regarding the noise levels, it seems that results for SNR = 5 dB or even SNR = 1 dB show fewer vertical and horizontal artifacts than those for SNR = 10 and SNR = 100. Could you explain this?

Additionally, for the initial model cases, could you clarify why the less smooth model (case 1) results in a higher NMSE (0.04) than case 2?

3. Comments in the Text

Since I have not received the updated file, I will address a few points:

- Line 95 to the end of page 3:

My comment was that the statement "The membership sections that show the degree of belonging of each data point to different rock types or facies are new concepts that could be used in seismic interpretation" is not a new concept.

Your response was, "I couldn't find any previous work that creates membership sections. Please see section 3."

However, this concept has been presented in several works, including Kieu and Kepic (2020), which is already in your reference list. I believe you need to review the previous work more thoroughly.

- Page 10, Line 298:

My question is "why do you use 6 clusters instead of 11 as the results of elbow method (line 277 and 278)?".

Your answer "(Chevitarese et al., 2018)"

This work is not referenced in the relevant statement within your manuscript. I think this is not a proper scientific explanation. It is your responsibility to provide a convincing, evidence-based justification for your choice.

- Page 23, Figure 5:

My comments "I think low resolution of the image make difficult to see. The caption of figure should detail enough to help reader understand major idea of the figure without reading the text".

Your answer "the original figure is a sharp 500dpi image. something happens while converting the files to pdf. We have replaced it again"

It is your responsibility to ensure the quality of all versions of your manuscript.

I look forward to your revised manuscript with these points addressed.

Best regards,

---

## Community Comment (CC1)

Dear Authors,

I believe your work has potential. However, there are several issues that need to be discussed in more detail and with greater clarity. I have included my comments in the text file.

Here are some main concerns:

How do you choose the parameters for the objective function (Equation 14), and what criteria do you use to select these parameters?

How do you choose the parameters for clustering, and what criteria do you use to select them?

Could you please provide results of the inversion with different noise levels and different starting models, and compare these results with those obtained using Hampson-Russell?

Kind Regards

=================================================================================

Dear Reviewer,

Thank you for your thorough review and insightful comments on our manuscript. We appreciate the time and effort you have invested in providing detailed feedback. Below, we address each of your observations and comments.

We hope the following information and explanations meet your expectations.

1. How do you choose the parameters for the objective function (Equation 14), and what criteria do you use to select these parameters?

Answer:
As stated in Equation 14, "Each objective term is weighted using a denominator, $\theta(\cdot)$, to normalize the contribution of the terms." The specific forms of $\theta$ are given by:

$$\theta(d) = \frac{norm(\boldsymbol{d}.2)}{length(\boldsymbol{d})}. \quad \theta(\boldsymbol{x}) = \frac{norm(\boldsymbol{x}.2)}{length(\boldsymbol{x})}. \quad \theta(c) = \frac{norm(\boldsymbol{u}^q\,\hat{O}.2)}{length(\boldsymbol{x}) \times C}$$

Additionally, lines 219-222 state, "Weighting factors balance the misfit or data fidelity term and the other terms (such that ($w\_d + w\_x + w\_s + w\_C = 1$)). This eliminates the need for high-dimensional L-curve, L-surface, or Generalized Cross-Validation methods to find optimum regularization parameters."

The weighting factors ($w\_d$), ($w\_x$), ($w\_s$), and ($w\_C$) can have equal contributions, or the operator can adjust them based on initial information about the model.

- - - - - - - - - - - - - - - - - - - - - - - - - - - - - - - - - - - - - - - - - - - - - - - - - - - - - -

2. How do you choose the parameters for clustering, and what criteria do you use to select them?

The fuzziness parameter (($q$)) controls the clustering result. As stated in Figure 8 and lines 322-323, "the lower fuzziness tends the fuzzy clustering toward hard clustering and decreases the resolution of the membership sections."

By default, $(q = 2)$. However, to create more distinct clustering membership sections (e.g., to distinguish main sequences), one could decrease $(q)$. Conversely, to create less distinct clustering memberships (e.g., to distinguish within-layer lithologies), one could increase $(q)$. Furthermore, the role of $(q)$ in the interpretation of the results could be an interesting topic for further research.

- - - - - - - - - - - - - - - - - - - - - - - - - - - - - - - - - - - - - - - - - - - - - - - - - - - - - - - - - - - -

3. Could you please provide results of the inversion with different noise levels and different starting models, and compare these results with those obtained using Hampson-Russell?

To maintain the simplicity of the manuscript and avoid presenting extra information and results, we prefer not to add other examples to the manuscript. However, here we present the requested results for different noise levels and different initial models:

[Figure]

| SNR | Initial Model | Hampson Russel | FSI | |
|-----|---------------|----------------|-----|---|
| 100db | NMSE=0.33 | NMSE=0.24 | NMSE=0.004 | |
| 10db | NMSE=0.33 | NMSE=0.25 | NMSE=0.006 | |
| 5db | NMSE=0.33 | NMSE=0.25 | NMSE=0.01 | |

[Figure]

[Figure]

| No. | Initial Model | Hampson Russel | FSI | |
|-----|---------------|----------------|-----|---|
| 1 | NMSE=0.15 | NMSE=0.12 | NMSE=0.04 | |
| 2 | NMSE=0.33 | NMSE=0.26 | NMSE=0.02 | |
| 3 | NMSE=0.39 | NMSE=0.37 | NMSE=0.05 | |

| | | | | |
|---|---|---|---|---|
| 4 |
[Figure]
NMSE=0.50 | NMSE=0.47 | NMSE=0.05 | |

Please note that for the inversion process using the less accurate initial model, we have reduced the weight of the initial model constraint ($w_m$)

- - - - - - - - - - - - - - - - - - - - - - - - - - - - - - - - - - - - - - - - - - - - - - - - - - - - - - - - - - - - - - - - - - - - - - - - - - - - -

Kind regards

---

## Author Comment (AC1)

[revised manuscript text omitted]

$$\boldsymbol{r}_j = \frac{1}{2}\ln\left(\frac{\boldsymbol{z}_{j+1}}{\boldsymbol{z}_j}\right) = \frac{1}{2}\log(\boldsymbol{z}_{j+1}) - \frac{1}{2}\log(\boldsymbol{z}_j) = \boldsymbol{x}_{j+1} - \boldsymbol{x}_j. \quad (a)$$

$$\boldsymbol{r} = \boldsymbol{D}_v\boldsymbol{x}. \quad (b)$$

( 2 )

where $\boldsymbol{Z}_j$ is the acoustic impedance of the $j^{\text{th}}$ sample point, and $\boldsymbol{D}_v$ is the first-order differential matrix along the time dimension. Conversely, having $\boldsymbol{r},$ the value of x can be obtained in a recursive calculation using Eq. ( 3 ):

$$\boldsymbol{x}_{j+1} - \boldsymbol{x}_j = 2\sum_{i=1}^{j}\boldsymbol{r}_i. \quad (a)$$

$$\boldsymbol{x} = \boldsymbol{H}\boldsymbol{r}. \quad (b)$$

$$\boldsymbol{z}_j = \boldsymbol{z}_1 e^{2\sum_{i=1}^{J} r_j} = \boldsymbol{z}_1 e^{\boldsymbol{H}\boldsymbol{r}}. \quad (c)$$

[revised manuscript text omitted]

---

## Author Comment (AC2)

Dear Authors,

Thank you for your responses and explanations. While I agree with some of your points, overall, I am not satisfied with the responses. I would like to address three key aspects: the format of the responses, the answers to my main questions, and the comments in the text.

Dear Reviewer

Thank you for your detailed and thorough review of the manuscript. We have addressed your previous comments, which have significantly improved the manuscript. For the remaining concerns, we have provided responses below, and we hope they convey our message clearly.

==============================================================================

1- The form of the answers.

I could not locate an updated version of your manuscript; I only see the old file with your responses. It is difficult to follow your answers without the use of track changes. Please ensure that the updated manuscript with track changes is provided for easier review.

**Answer**: I am aware of that. Unfortunately, there isn't an option to upload the modified manuscript at this time. I have contacted the journal, and their response was to wait for the option to upload the revised manuscript as a formal response to the comments. Since the review process is anonymous, I cannot send the file to you directly. I hope the upload option becomes available soon, allowing us to continue the review process smoothly.

==============================================================================

2- The answers of my "main concerns"

a) Objective Function Parameters (Equation 14):

You stated that the weighting factors ($w_d$, $w_x$, $w_s$, $w_C$) either have equal contributions or are adjusted based on initial model information. My concern is that if these parameters have no effect when equal ("equal contributions"), how does the operator determine their weighting values?

**Answer:** Added to the manuscript " The proposed weighting parameters have eased the process of determining optimal regularization parameters (Eq. (14)). By default, all weights are set equally, but the operator can adjust them according to the reliability of each respective term. For instance, when seismic data has a higher SNR, the weight assigned to the data fidelity term can be increased. This adjustment also applies to the initial model and prior clustering data. "
* * *
b) Could you please provide a clearer explanation of how prior information (e.g., geology, boreholes) is incorporated into the inversion process? I suggest adding more detailed statements, discussion, and supporting evidence to the text.

**Answer:** Prior information

- **Initial Model:** In model-based inversion methods, an initial guess helps constrain the solution space, limiting possible models to those that are close to the initial estimate, which is crucial given the ill-posed nature of the inversion problem.
- **Initial interpretation of seismic sections and geological data (if available):** Preliminary interpretations, such as identifying general layers and sequences, can be used as initial or fixed clustering data.
- **Well-log data:** Well-logs provide the primary source for initial or fixed clustering data. In the examples, fuzzy c-means (FCM) clustering of acoustic impedance (AI) derived from density and sonic logs was used to determine both the number of AI clusters and the value of each cluster.

The manuscript discusses various scenarios and approaches, exploring several possible sets of initial FCM data.
* * *
c) Clustering Parameters: The parameters related to clustering include the "Gustafson-Kessel fuzzy C-means" method, fuzziness, and the number of clusters.

- Your theory section discusses conventional Fuzzy C-means (FCM), but there is no clear distinction between FCM and the Gustafson-Kessel method in relation to your inversions.

**Answer:** There are two different use of clustering in the manuscript. First, FCM objective function within the proposed inversion objective function, Second, in the clustering of well-log data.

The first one is explained in one sub-section.

The second one is a common clustering process. Although, it is not necessarily required, for the second purpose we have employed the Gustafson-Kessel. Gustafson-Kessel is adaptable to clusters of various forms and sizes. This versatility is critical in geophysical applications, since subsurface features are frequently complicated and irregular.

The main difference of FCM and GK is the distance calculation. This is mentioned in the theory section " … distances can be in various forms such as Euclidian (Eq. (9)) as in FCM or Mahalanobis (Mahalanobis, 1936) as in Gustafson Kessel (GK) fuzzy clustering (Krishnapuram and Kim, 1999)."

To address your concern another sentence is added to the manuscript " Although, it is not necessarily required, we preferred to use Gustafson-Kessel in the clustering of the well-logs. Gustafson-Kessel is adaptable to clusters of various forms and sizes. This versatility is critical in geophysical applications, since subsurface features are frequently complicated and irregular. "
* * *
- The theory section lacks a discussion of fuzziness and the number of clusters.

**Answer:** Added "where q is a parameter that indicates the fuzziness of the data, which controls the degree of cluster overlap, with higher values of q leading to more overlap and softer clustering boundaries. C represents the number of clusters within the data."
* * *
- Although spatial properties are mentioned, I do not see them addressed in the inversion process.

**Answer:** Thank you for noting that. The manuscript is edited "… here these updates happen just once per each iteration of the inversion using Eq. (10) and Eq. (14)."
* * *
Overall, this research employs a variety of methods and concepts. In a research paper, we assume that readers possess a general understanding of the concepts utilized. Our primary focus is to present the proposed method. Providing detailed overviews of concepts such as optimization, the elbow method, Tikhonov optimization, clustering methods, clustering parameters, optimization techniques for clustering, …., which are well-established, would significantly lengthen the manuscript. This information is readily available in numerous books and articles.

===================================================================================

Please clarify these points and provide relevant discussions and evidence in the manuscript.

- Inversion Results with Different Noise Levels and Starting Models: I believe adding results for different noise levels and initial models would not require much extra space if you manage the presentation more concisely and reduce the number of cases.

**Answer:** Thank you for your concern. We have presented the results for a nearly worst-case scenario of signal-to-noise ratio (SNR) in practical applications. Given the pre-print policy of the journal and the online availability of comments and responses, we prefer not to complicate the manuscript further. The manuscript already incorporates various concepts, and an SNR of 1 dB is considered an acceptable result.
* * *
- Regarding the noise levels, it seems that results for SNR = 5 dB or even SNR = 1 dB show fewer vertical and horizontal artifacts than those for SNR = 10 and SNR = 100. Could you explain this?

**Answer:** Based on your previous concern regarding the weighting parameters and the operator's ability to select them based on the reliability of seismic data, the initial model, and the initial clusters: in cases of noisy data, the operator can increase the weight for smoothness and decrease it for data fidelity. This adjustment has resulted in greater lateral smoothness in the results for higher SNR. However, this approach comes at the expense of model values and resolution, and it does not imply that we must apply higher weights for smoothness in high SNR scenarios.

Also, since noise is introduced randomly, a small difference in the final model is expected.
* * *
Additionally, for the initial model cases, could you clarify why the less smooth model (case 1) results in a higher NMSE (0.04) than case 2?

**Answer:** Thank you for pointing that out. As shown in the fourth column, the first case represents the initial model, which is closer to the true model, resulting in smaller NMSE values for both the initial and final models. The NMSE for this case was actually 0.004, which I rounded up to 0.01. However, it was mistakenly listed as 0.04 in the table.

========================================================================

3- Comments in the Text

Since I have not received the updated file, I will address a few points:

- Line 95 to the end of page 3: My comment was that the statement "The membership sections that show the degree of belonging of each data point to different rock types or facies are new concepts that could be used in seismic interpretation" is not a new concept.

Your response was, "I couldn't find any previous work that creates membership sections. Please see section 3." However, this concept has been presented in several works, including Kieu and Kepic (2020), which is already in your reference list. I believe you need to review the previous work more thoroughly.

**Answer:**

(a) Membership sections are vertical profiles that represent the degree of belonging of subsurface data points to individual clusters. If there are $C$ clusters, there will be $C$ separate membership sections.
(b) Membership sections, which are a new concept and a natural output of fuzzy inversion, are referred to as 'membership degree' in Kieu and Kepic (2020). These sections have significant potential as an aid in geological interpretation.
(c) Kieu and Kepic (2020) impose geological unit boundaries on the clusters, which biases the clusters toward these predefined units. So, despite its promising potential, I am uncertain about how to effectively utilize the 'pseudo-lithology' data produced in their research. However, this topic could be explored in a comprehensive study on the use of 'membership sections' and 'pseudo-lithology' in various geological and petrological scenarios, alongside other geophysical data and seismic attributes."
* * *
- Page 10, Line 298: My question is "why do you use 6 clusters instead of 11 as the results of elbow method (line 277 and 278)?". Your answer "(Chevitarese et al., 2018)"

This work is not referenced in the relevant statement within your manuscript. I think this is not a proper scientific explanation. It is your responsibility to provide a convincing, evidence-based justification for your choice.

**Answer:** It is stated in the first paragraph of section 3.2 Real case example " The elbow method for Gustafson-Kessel clustering of impedance well log suggests 11 clusters in the sampled depth. However, the seismic

horizons show six general facies segments (Chevitarese et al., 2018) and some sublayers." Both of these cluster numbers are tested.
* * *
- Page 23, Figure 5:

My comments "I think low resolution of the image make difficult to see. The caption of figure should detail enough to help reader understand major idea of the figure without reading the text". Your answer "the original figure is a sharp 500dpi image. something happens while converting the files to pdf. We have replaced it again" It is your responsibility to ensure the quality of all versions of your manuscript.

**Answer:** Thank you for pointing out the issue. I have replaced the figure.

=====================================================================================

Once again, we appreciate the time and effort you have dedicated to reviewing our manuscript. We have carefully considered and addressed each of your comments, which have helped improve the overall quality of the work. We hope that our responses and the corresponding revisions meet your expectations and clarify any remaining concerns.

We hope the link to upload the revised manuscript will be activated soon, allowing us to share the updated version with you. Otherwise, you may request the file directly from the editorial office!

Thank you for your valuable feedback.

---

## Author Comment (AC5)

Dear Reviewer,

Thank you for your review and insightful comments on our manuscript. We sincerely appreciate the time and effort you have dedicated to providing such valuable feedback. Below, we have addressed each of your observations and comments in detail. We hope that our responses and explanations meet your expectations and clarify any concerns.

========================================================================

**RC:**

The research in this article, which uses fuzzy clustering optimization for wave impedance inversion, belongs to a relatively outdated approach. The principle description of the paper is quite detailed, but most of it belongs to the derivation of classical theories. However, the following concerns need to be addressed before the paper can be accepted for publication.

**Answer:**

The proposed algorithm introduces a custom definition of fuzzy clustering optimization as an additional term in the objective function for acoustic impedance inversion. The approach, its solution, and the resulting outcomes are novel, demonstrating the potential of the method. Therefore, we respectfully disagree with the characterization of the proposed approach as 'outdated.' It seems that this term may instead be referring to the individual optimization terms included in the objective function. However, these terms are widely utilized in recent related research. The proposed algorithm combines the advantages of these terms while mitigating their cumulative drawbacks. Furthermore, modifications and improvements have been implemented to address the detailed comments provided by the editors.

- - - - - - - - - - - - - - - - - - - - - - - - - - - - - - - - - - - - - - - - - - - - - - - - - - - - - - - -

1. Algorithm flow or algorithm pseudocode should be supplemented;

   **Answer:**

   Algorithm 1 is added to explain the propose method.

- - - - - - - - - - - - - - - - - - - - - - - - - - - - - - - - - - - - - - - - - - - - - - - - - - - - - - - -

2. The visual images provided by the author, such as seismic profiles and inversion results, have poor drawing quality;

   **Answer:**
   The figures have been revised and provided with a resolution of 500 dpi. Additionally, the figures are available as separate files.

- - - - - - - - - - - - - - - - - - - - - - - - - - - - - - - - - - - - - - - - - - - - - - - - - - - - - - - -

3. The evaluation of inversion results is not very scientific and reasonable. It is recommended to compare and quantitatively evaluate them with logging data.

**Answer:**

Figure 7 and Table 1 evaluate the results of the inversion and compare them with the available well log data.

- - - - - - - - - - - - - - - - - - - - - - - - - - - - - - - - - - - - - - - - - - - - - - - - - - - - - - - - - - -

4. The conclusion section needs to discuss the limitations of the method.

**Answer:**

Thank you for noting this issue. The conclusion section is revised accordingly.